# Exceptionally low genomic diversity in the underutilised legume Kersting's groundnut

Tsz-Yan Cheung [1,2], Konoutan M. Kafoutchoni [3], Eric E. Agoyi[3], Ting-Fung Chan [1,2] & Mark A. Chapman [4] ✉

Identifying crops with novel and climate resilience traits is imperative to ensure food security. Without a basic understanding of the genomes and genomic diversity of these crops they will remain underutilised or could even become lost. Kersting's groundnut [*Macrotyloma geocarpum* (Harms) Maréchal & Baudet] is one such crop, regarded as a useful, drought tolerant and sometimes valuable legume. Here, we present the assembly and annotation of the genome of Kersting's groundnut and an analysis of genomic diversity across a diversity panel. Accessions are grouped by geography and seed coat colour, one of the key traits used to describe the accessions. Four candidate genes involved in pathways relating to pigments or flavonoids are revealed. One of the important findings is that Kersting's groundnut retains very low diversity, about 85-95% less than two other legumes, suggesting a pressing need to conserve the existing diversity of Kersting's groundnut.

Achieving food and nutrition security is an international challenge which entails growing more food, on less land, in an increasingly unpredictable and hostile climate[1–3]. Crop breeding has largely kept up with climate change, however, recent climate events have challenged even the most robust modern crop varieties, with future yields predicted to decline[4,5]. Whilst breeding staple crops for a future climate is a major target, expanding the repertoire of crops we rely on can act in concert. This would entail identifying crops that are suited to the location and (future) climate, and there are dozens of underutilised crops which could fulfil this role[6–11].

Underutilised crops represent a subset of minor crop species that demonstrate untapped potential[12]. This potential often comes in the form of abiotic tolerance, for example, if they are native to hot and dry locations, then they are likely to cope with the future climate. They may be locally grown as an insurance crop, likely to yield when harsh weather causes a staple crop to fail, and may play a role in traditional foods or ceremonies. Their extreme adaptation means they are useful models for understanding adaptation to harsh conditions and could serve as donors of adaptive alleles either through breeding (if they are closely related to a more widespread crop) or via genetic modification

approaches. Further, these crops can bring novel or enhanced nutritional attributes which would benefit the consumers.

However, there are reasons these species have not made it to international recognition, for example, containing anti-nutrients, having significant harvesting or growing difficulties and/or being pest and pathogen susceptible[11]. Some underutilised crops are at risk of becoming lost when seeds and the associated indigenous knowledge are not passed down[13]. It is imperative to understand and maintain the genetic variation in these (and all) crops to ensure future breeding efforts to improve these crops are as efficient as possible[14].

Several underutilised crops have achieved recognition since the 1970s; however, with one or two exceptions, expansions into the global markets are yet to take place. Expansion has been successful for crops such as quinoa and pigeonpea, which, until the late twentieth century, were minor, locally grown crops. More recently, several reviews have popularised the use of underutilised crops, and we are beginning to understand more about their tolerances, genomes and genomic variation. With this information, it is possible to identify, examine and breed these crops, as well as to use the information from these crops to determine future priorities. Recently, for example, novel crops for the

[1]School of Life Sciences, The Chinese University of Hong Kong, Hong Kong SAR, China. [2]Center for Soybean Research of the State Key Laboratory of Agrobiotechnology, The Chinese University of Hong Kong, Hong Kong SAR, China. [3]Non-Timber Forest Product and Orphan Crop Species Unit, Laboratory of Applied Ecology, Faculty of Agronomic Sciences, University of Abomey-Calavi, Abomey-Calavi, Benin. [4]School of Biological Sciences, University of Southampton, Southampton, UK. ✉e-mail: m.chapman@soton.ac.uk

future of UK agriculture were prioritised for investigation, resulting in a shortlist of ~30, including chickpea, lentil, buckwheat and sunflower[15], with similar initiatives in South Africa[16] and Ghana[17].

Here we focus on the underutilised legume Kersting's groundnut (*Macrotyloma geocarpum* (Harms) Maréchal & Baudet) because recent studies have revealed its nutrition and acceptability[18,19], constraints on production and wider adoption[20–22], and its genetic variation[23,24]. These are all vital research avenues which can together help promote this crop; however, to expedite breeding and the understanding of the genetic basis of adaptive traits, a genome sequence and an in-depth understanding of genomic variation would be of substantial benefit[25]. So far, no genome sequence has been generated except for a fragmented genome generated with short-read sequencing that was used for identifying genetic markers[26].

Kersting's groundnut is grown in West Africa, primarily Burkina Faso, Ghana, Togo, Benin and Nigeria. It has a high protein content (20–22%) and is relatively high in iron, zinc, calcium and magnesium[19,27] and has been explored as a mechanism to improve the protein content of snacks by incorporating Kersting's groundnut flour into the mix[28]. It is preferred locally over other more readily available beans (cowpea and Bambara groundnut)[18], which contributes to its value as a crop. It is adapted to areas of low rainfall and cannot survive long periods of excess rainfall[18].

A recent survey in Benin showed that Kersting's groundnut is prone to fungal diseases and insects, which may be a constraint on production, but no control strategies are employed[22]. A lack of improved cultivars and insufficient seed availability in some years are also cited as reasons constraining production[29], and opinions on the uses, cost, husbandry and constraints vary between parts of the distribution throughout which Kersting's groundnut is grown[30]. Cooking time can be an hour or more[19]; several underutilised legumes suffer from long cooking times, limiting their widespread use[31]. Identifying candidate genes and quantitative trait loci could help to expedite future breeding programmes, but for many underutilised crops, the genetic basis of adaptive traits is poorly understood[25].

Phenotypic variation in Kersting's groundnut is low; in general, varieties are differentiated by seed coat colour alone, or in combination with leaf shape in some regions[20,30,32]. This is coupled with low genetic variation; an early study using isozymes identified no variation among 32 putative loci in 18 accessions of Kersting's groundnut[33]. Recent work using high-throughput sequencing has allowed the identification of hundreds of SNP loci across large collections of phenotyped accessions (>200), yet still diversity is low[23,24]. Nonetheless, these studies identified differentiation among populations and accessions which matches geographic and morphological groups. Studies of genetic variation in Kersting's groundnut, such as these, have typically identified genetic differentiation between the North-Western part of the distribution (Sudanian region) and the South-Eastern part (Guinean region).

Given the research above indicates that genetic variation is very low in Kersting's groundnut, it is imperative to understand how variation is partitioned and to ensure that adaptive variants are not lost. A valuable addition to the existing repertoire of information on this crop, which could enable future molecular breeding efforts, would be the development of an annotated reference genome.

In this work, we generate an annotated reference genome and resequence a panel of accessions to quantify genome-wide variation and to identify how the accessions group or are differentiated. We find that accessions with the same seed colour are genetically grouped and use this to determine if we can discern the genetic basis of seed colour in Kersting's groundnut.

## Results
### Genome assembly
To assemble a high-quality genome for Kersting's groundnut, we generated 1.87 M PacBio HiFi reads (30.3 Gb) with a N50 of 16.4 kb,

**Table 1 | Summary of genome assembly and annotation**

| Category | Metric | Value |
|---|---|---|
| Assembly | Accession | Tkg-36 |
| | Assembly size in bp | 365,532,019 |
| | Number of scaffolds | 1312 |
| | Scaffold N50 in bp | 30,590,648 |
| | L50 | 6 |
| | Longest scaffold in bp | 39,513,689 |
| | GC content | 32.46% |
| | No. of gaps | 231 |
| | No. of *N* in bp (% of genome) | 115,500 (0.032%) |
| | Complete BUSCOs | 98.3% (S: 94.5%, D: 3.8%) |
| | Average LTR Assembly Index (LAI) | 17.2 |
| Gene prediction | No. of predicted genes | 33,193 |
| | No. of transcripts | 35,550 |
| | No. of tRNA | 4866 |
| | No. of lncRNA | 6714 |
| | Complete BUSCOs | 92.8% (S: 76.8%, D: 16.0%) |
| Repeat elements | LTR in bp (% of genome) | 65,089,093 (17.81%) |
| | LINE in bp (% of genome) | 2,121,751 (0.58%) |
| | SINE in bp (% of genome) | 2,944,038 (0.81%) |
| | DNA transposons in bp (% of genome) | 7,460,237 (2.04%) |
| | Rolling circle in bp (% of genome) | 2,342,530 (0.64%) |
| | Others in bp (% of genome) | 33,231,225 (9.10%) |
| | Unclassified repeat in bp (% of genome) | 72,150,883 (19.74%) |
| | Total repeat in bp (% of genome) | 185,339,757 (50.70%) |

representing 77.5× coverage of the Kersting's groundnut genome based on the previously estimated genome size of 391 Mb[34]. The initial genome was assembled into 1044 contigs, with a total size of 365.4 Mb and a contig N50 of 31.4 Mb. Through incorporating 26.6 Gb of Omni-C data, the genome was further polished by correcting and orienting the contigs into pseudomolecules, generating a scaffolded assembly in 1312 scaffolds with a total size of 365.5 MB and N50 of 30.6 Mb (Table 1). Given the high contiguity and N50 of the contig-level assembly, Omni-C scaffolding resulted in only marginal improvement. Although the incorporation of Omni-C data improved the accuracy of the genome assembly in terms of structural and spatial organisation by splitting and reorienting contigs, it led to an increase in scaffold count. In total, 83.8% (306.4 Mb) of scaffolds are anchored into 10 pseudo-chromosomes, corresponding to the haploid chromosome number of Kersting's groundnut (Fig. 1a, b). The assembled chromosomes were mined for microsatellites (simple sequence repeats) for potential future use (Supplementary Data 1; see 'Methods'). The majority of microsatellite motifs were mononucleotides (68.5%) with di-, tri- and tetranucleotide repeats making up 19.2%, 4.8% and 5.4% of the motifs. Penta- and hexanucleotide repeats collectively made up the remaining 2.1% of motifs.

### Genome annotation
A total of 434,309 repetitive sequences were identified in the assembled genome, comprising 185 Mb (50.70% of the genome) with 207,232 elements (19.74% of the genome) being unclassified (Fig. 1c; Table 1). Among the classified repeats, long terminal repeat retrotransposons (LTR-RT) under class I transposable elements

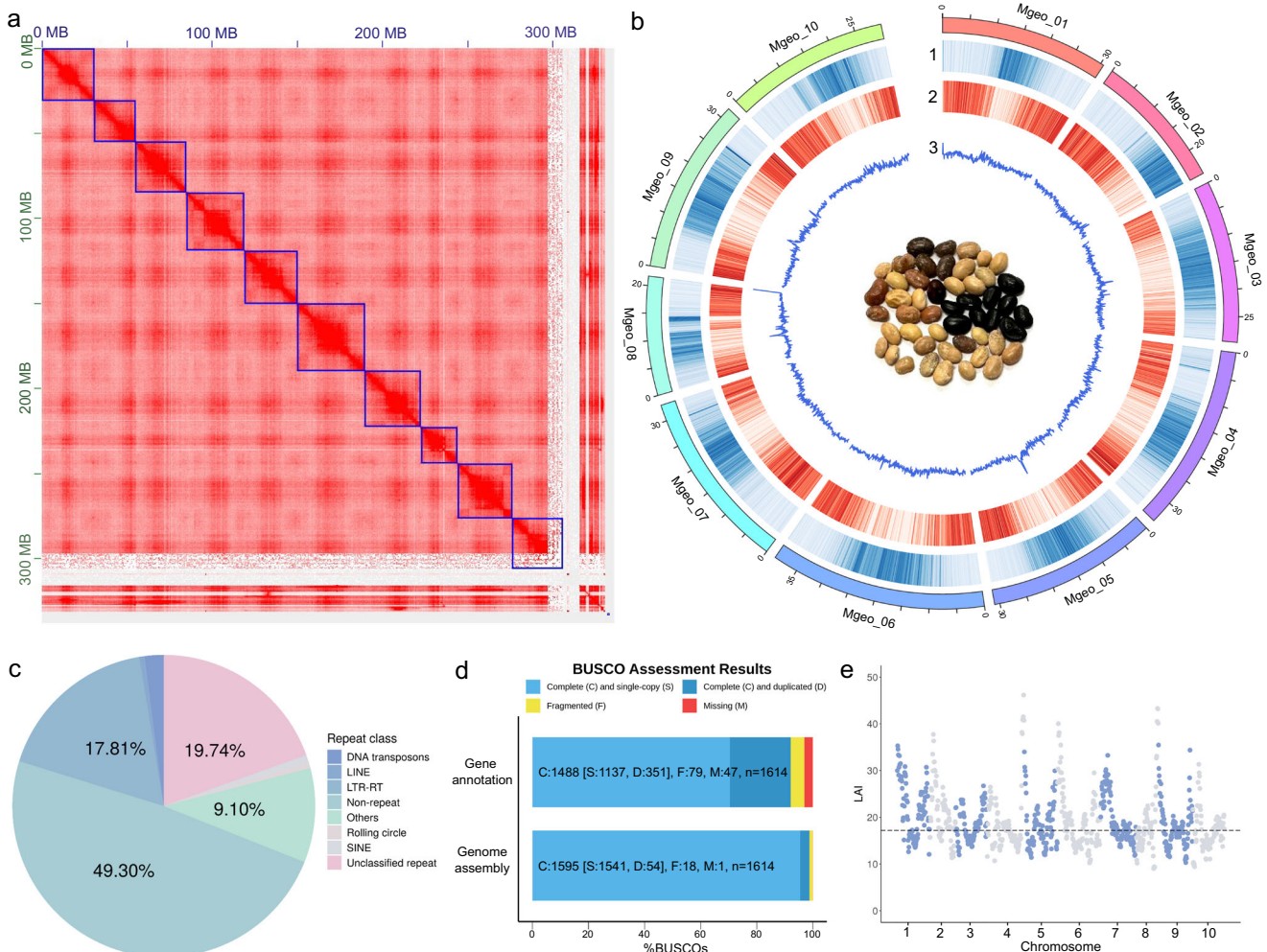

**Fig. 1 | Genome assembly of *M. geocarpum*. a** Omni-C contact map of the genome assembly visualised using Juicebox. **b** Distribution of genomic features of the assembled genome. The outer layer illustrates the 10 pseudochromosomes with length in megabases (Mb). The tracks from the outer to the inner track show (1) repeat density, (2) gene density and (3) GC content calculated in a 100 kb window. Photo taken by T.-Y. Cheung. **c** Distribution of repetitive elements in the assembled genome. **d** BUSCO assessment of the genome assembly and gene annotation using the embryophyta lineage. **e** LTR Assembly Index (LAI) distribution of the 10 pseudochromosomes. The dashed horizontal line represents the average whole-genome LAI value. Source data are provided as a Source Data file.

were the most abundant type of repetitive elements, occupying 65.1 Mb (17.81% of the genome). In contrast, class II transposable elements (DNA transposons) only comprised 2.04% of the genome, including 0.38% of hobo-Activator and 0.10% of Harbinger. Among the transposable elements, Gypsy occupied the greatest proportion of the genome (20,716 elements, 22.6 Mb [6.18%]), followed by Copia (20,239 elements, 16.4 Mb [4.48%]). Long interspersed elements (LINEs) and short interspersed elements (SINEs) occupied 0.58% and 0.81% of the genome, respectively. In addition, small RNAs occupied a significant proportion in the genome compared to transposable elements, occupying 24.0 Mb (6.55% of the genome assembly).

To predict protein-coding genes, we generated 16.6 Gb Illumina RNA-seq reads and 856 Mb (1.03 M reads) Nanopore direct RNA sequencing reads from leaf tissue. Using both the short- and long-read transcriptomic data as transcript evidence, we predicted a total of 33,193 gene models with 35,550 protein-coding transcripts in the genome of Kersting's groundnut (Table 1). Among the identified transcripts, 33,240 transcripts (93.5%) encode proteins that have predicted functional domains. For the non-coding RNA, 4866 tRNAs were identified, encoding the standard 20 types of amino acid, while 6714 lncRNAs were predicted.

## Evaluation of genome assembly

Benchmarking Universal Single-Copy Orthologs (BUSCO) assessment illustrated a high level of completeness in the genome of Kersting's groundnut. 98.3% (with 94.5% as single-copy gene) and 92.8% (with 76.8% as single-copy gene) of the BUSCOs were complete in the genome assembly and gene annotation, respectively (Fig. 1d; Table 1). Additionally, a whole-genome LTR Assembly Index (LAI)[35] of 17.2 was reported, based on the content of intact LTR-RTs, with nearly all chromosomal regions exhibiting a regional LAI > 10, indicative of a reference-grade genome assembly[35] (Fig. 1e). To further evaluate the completeness of the genome, pseudochromosomes of Kersting's groundnut were compared against two other legumes, *Sphenostylis stenocarpa* (African yam bean) and *Vigna angularis* (adzuki bean), to examine the homologous chromosomal regions and pairwise syntenic gene blocks based on their gene annotation. Despite the differences in genome size and a difference in chromosome number among the three species, the syntenic comparison displayed large blocks of homology between the 10 pseudochromosomes of Kersting's groundnut and the other legume genomes with several inter-chromosomal rearrangements (Fig. 2). *M. geocarpum* has ten pairs of chromosomes, whereas *S. stenocarpa* and *V. angularis* have 11, despite *M. geocarpum* and *S. stenocarpa* being

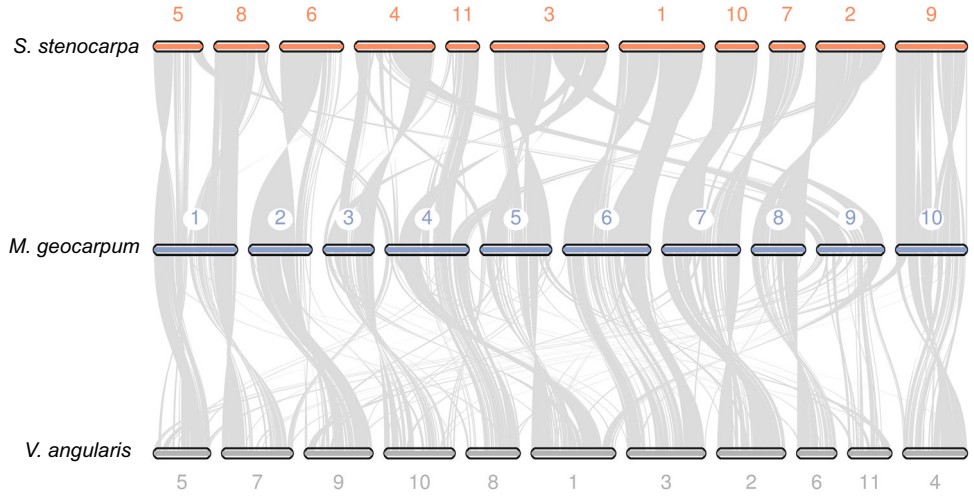

**Fig. 2 | Syntenic comparison of the pseudochromosomes between *S. stenocarpa*, *M. geocarpum* and *V. angularis*.** The chromosomes of *S. stenocarpa* and *V. angularis* were reoriented for visualisation.

more closely related to each other[36]. This difference largely comes from *M. geocarpum* chromosome 1 being syntenous with two chromosomes in the other species. Few chromosomes demonstrate complete 1:1 synteny, but no large-scale synteny blocks were missing among the three genomes, suggesting that the core genomic structures and conserved gene blocks were preserved in the genome of Kersting's groundnut at a high degree of completeness.

### Gene family analysis

Gene families in Kersting's groundnut were identified by comparing the 28,327 protein sequences with four other legumes, namely *Medicago truncatula* (barrel medic), *Phaseolus vulgaris* (common bean), *Pisum sativum* (field pea) and *Vigna angularis*, using *Arabidopsis thaliana* as the outgroup. In total, 23,505 Kersting's groundnut genes (83.0%) were assigned to 16,656 gene families, with the remaining 4822 unassigned. Comparing the gene families in the five legume species, 13,751 gene families were shared by all legumes, while 56 genes in 26 gene families were unique in Kersting's groundnut (Fig. 3a). Superimposing gene families onto the phylogenetic relationship of the five legumes and *Arabidopsis*, we identified 701 expanded and 1249 contracted gene families in Kersting's groundnut (Fig. 3b). Of these, 97 gene families were significantly ($p < 0.05$) expanded and 243 gene families were significantly contracted. We assigned gene ontology (GO) terms to those genes in the significantly expanded and contracted gene families and performed GO enrichment analysis. This yielded 84 and 33 significantly enriched GO terms (adjusted $p < 0.05$) in the lists of expanded and contracted gene families, respectively. Expanded gene families were mainly enriched for terms related to photosynthesis and energy production, for instance, electron transport in photosystem, chlorophyll binding and ATP biosynthetic and metabolic processes. For the contracted genes, GO terms related to a wide range of biological processes and functions, including photosynthesis and respiration, transcription, and biosynthesis of secondary metabolites, were enriched (Fig. 3c, d).

### Resequencing, phylogeny and genetic diversity

We obtained 46.1 M (±3.3 M [S.D.]) paired-end reads per sample, and after trimming 44.6 M (±3.2 M) were retained (Supplementary Data 2). The percentage of reads mapping was high for all samples (95.1 % ± 3.5 %) and depth was 25.6 (±3.1; Supplementary Data 2). Depth was lowest for the outgroup, but still an average of 17.8 reads.

Including the outgroup, 5,007,665 polymorphic sites were identified (4,907,204 SNPs, 100,461 indels). Sites with >3 samples missing a call and indels were removed, and subsequently SNPs with a MAF < 0.05 were left, leaving 408,066 SNPs. After removing further SNPs due to LD, 162,490 were retained for phylogenetic analysis.

The NJ tree rooted with *M. stenophyllum* demonstrates that samples cluster by geography, with all samples from the same country (Ghana, Burkina Faso, Benin and Nigeria) forming monophyletic groups with 100% bootstrap support (BS) (Fig. 4). The samples split into two groups, with those from Ghana and Burkina Faso in one group and those from Benin and Nigeria in the other, i.e. demonstrating an east-west split. Limited phylogenetic structure is apparent within each group, and branches are often poorly supported with low BS values; however, within Nigerian samples, there is a robust split into two groups, four accessions from central and southwest Nigeria form one group, and six from the southeast form the other (Fig. 4).

For the population genomic analysis, after excluding the outgroup, 539,800 polymorphic variants remained (469,291 SNPs, 70,509 indels). Further reduction of the dataset by removing indels, sites with missing data and a low MAF and sites in LD (see 'Methods'), retained 156,650 SNPs for population genomic analysis with STRUCTURE. Analysis suggested that the optimal number of populations (*K*) was 4, but this provided no more resolution than *K* = 3 (Fig. 4). The STRUCTURE results parallel those in the phylogeny—samples from Ghana and Burkina Faso form one cluster, samples from Benin form a second, and the southeastern Nigerian samples form a third. The samples from central and southwest Nigeria appear intermediate between the Benin cluster and the cluster of the other Nigerian samples. Again, this supports that the genome-wide variation is strongly based on the geography of the samples.

Genetic variation was low in Kersting's groundnut despite our sequencing panel being geographically and morphologically diverse (see 'Methods'). The full dataset (i.e. after removing the outgroup but before removing sites with missing data, rare alleles and sites in LD; 469,291 SNPs and 70,509 indels) equates to 1 SNP and 1 indel every 653 bp and 4346 bp, respectively. Using the same settings, an analysis of 26 domesticated lablab samples and 18 cowpea samples reveals considerably greater diversity in both lablab (1 SNP and 1 indel every 34 and 277 bp) and cowpea (1 SNP and 1 indel every 77 and 648 bp). Taking the variants together, Kersting's groundnut therefore has ca. 1/18 the diversity of lablab and 1/8 the diversity of cowpea.

The three clusters described above (with all Nigerian samples in the third population, i.e. including the four with potentially intermediate genotypes) have low but similar diversity. Cluster 1 (Ghana

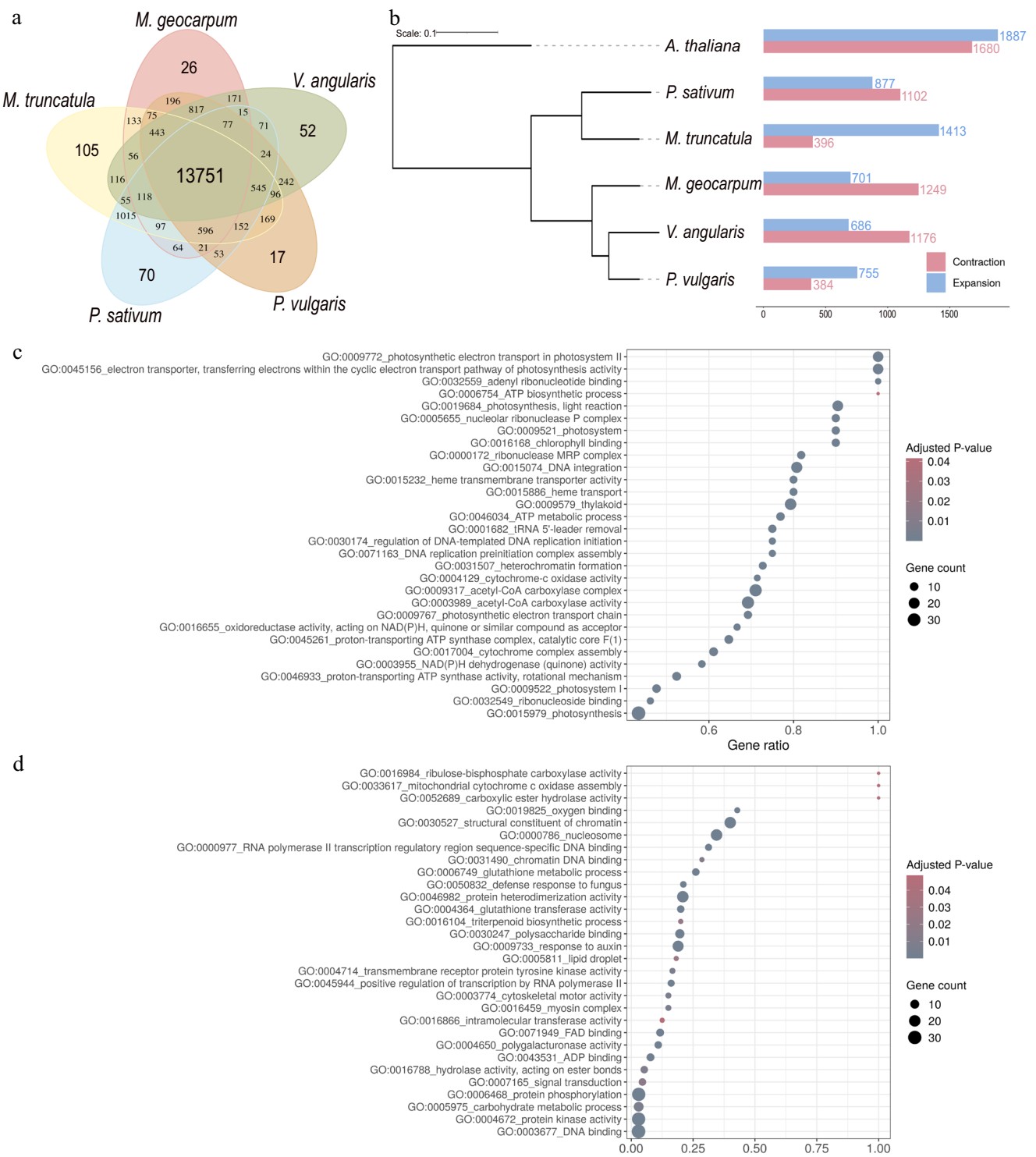

**Fig. 3 | Gene family analysis of Kersting's groundnut. a** Venn Diagram of the shared and unique gene families among Kersting's groundnut and four selected legume species (*Medicago truncatula, Phaseolus vulgaris, Pisum sativum* and *Vigna angularis*). **b** Phylogeny of Kersting's groundnut and the selected species with the number of expanded and contracted gene families in each species. The phylogenetic tree was constructed with iTol[98]. Gene ontology terms (adjusted $p < 0.05$)

enriched in the set of significantly expanded (**c**) and contracted (**d**) gene families in Kersting's groundnut using the one-sided over-representation analysis (hypergeometric test) for gene set enrichment. $p$ values were computed using Fisher's exact tests, and adjusted $p$ values calculated using the Benjamini–Hochberg method. Source data are provided as a Source Data file.

and Burkina Faso), cluster 2 (Benin), and cluster 3 (Nigeria) have genetic diversity (mean $\pi$ per 100 kb) of $0.570 \times 10^{-3}$, $0.501 \times 10^{-3}$ and $0.498 \times 10^{-3}$, respectively, and the mean number of SNPs per 100 kb bin is 144.4, 128.3 and 130.8, respectively. $F_{ST}$ was greater between clusters 1 and 2 (0.132) and clusters 1 and 3 (0.123) than between

clusters 2 and 3 (0.074), reflecting the closer similarity between clusters 2 and 3 in the population genomic analysis (Fig. 4).

We estimated $F_{IS}$, the inbreeding coefficient, which was high for all samples (0.52–0.58; Supplementary Data 2) and generally higher in clusters 1 and 2 than in cluster 3.

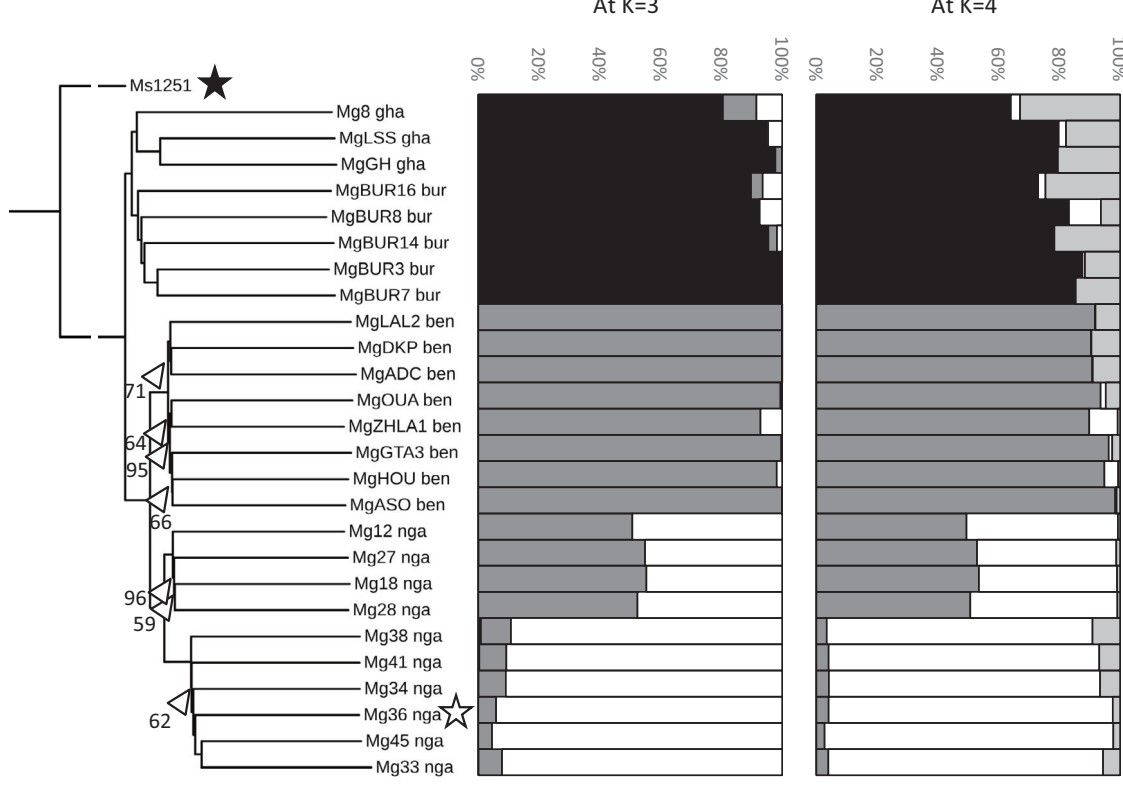

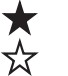 *M. stenophyllum* (outgroup)

☆ Sample for reference genome

**Fig. 4 | Phylogenetic and population genomic analysis of Kersting's Groundnut.** (Left) Neighbour Joining phylogenetic tree based on 162,490 genome-wide SNPs rooted with *M. stenophyllum*. Locations from where the accessions were collected are abbreviated as ben (Benin), bur (Burkina Faso), gha (Ghana), nga (Nigeria). Bootstrap values were 100% unless indicated. (Centre and Right) STRUCTURE analysis using $K$ (the number of putative populations) = 3 and 4. K = 4 was best supported, but gives the same resolution as $K = 3$. Individual accessions are represented as a horizontal bar, and membership to each of the 3 or 4 clusters is indicated by a different colour/shade. Source data are provided as a Source Data file.

## Candidate genes for seed colour

Four clusters of samples which shared the same seed colour were identified in the population genomic analysis. The three samples from Ghana plus two from Burkina Faso (BUR16 and BUR8; Fig. 4) formed the first black/brown group, sister to the remaining Burkina Faso samples, which all had white seeds with black spots. The Benin samples were all cream coloured, and the Nigerian samples were all brown or dark red. For both comparisons, the windows with the top 1% of $F_{ST}$ were identified, and the surrounding region which showed the top 5% of $F_{ST}$ was extracted and cross-checked between the two light-dark seed comparisons.

For the first comparison (five dark vs three white with black spots) the top 1% of $F_{ST}$ windows had $F_{ST} > 0.7790$ and ten regions of putatively high divergence were identified on chromosomes 1 (1 region), 2 (3), 3 (2), 7 (1), 9 (1) and 10 (2). For the second comparison (eight cream vs ten brown or red samples), the top 1% of $F_{ST}$ windows had $F_{ST} > 0.4702$ and 43 regions of putatively high divergence, at least one on every chromosome. None of the windows in either comparison overlapped with putative orthologues of the *Phaseolus vulgaris P* gene for white seed coat (Phvul.007g171300 and Phvul.007G171400); both are parts of the same gene[37]; this was identified on chromosome 7 (positions 18,230,603–18,249,975).

We cross-checked the results from the two comparisons and identified three regions that were in the top 1% of $F_{ST}$ in both comparisons, a further nine regions in the top 1% of $F_{ST}$ for one comparison and top 5% of the other, and a further 11 regions that were in the top 5% of $F_{ST}$ of both comparisons (i.e. they were flanking regions of the top 1% $F_{ST}$ windows, but not in the top 1% themselves). We searched for orthologues of genes in these regions and identified four with potential roles in pigmentation (Table 2).

## Discussion

Genome sequences can advance investigations of underutilised crops by aiding, for example, investigations into the genetic basis of traits, identifying adaptive/untapped diversity and solving taxonomy, e.g. identifying crop-wild relatives[12,25,38]. Reference genomes have been developed for a range of underutilised crops (teff[39], finger millet[40]), including many legumes (lablab[41], cowpea[42], winged bean[43], pigeonpea[44] and grasspea[45]).

Underutilised crops, due to advances in other crops and changes in cultures or preference, can face genetic erosion in their native location, resulting in the loss of alleles, varieties, or an entire crop species[46]. The 1960s green revolution saw the replacement of landraces with high-yielding inbreds, coupled with loss of habitat for crop-wild relatives[14]. Loss of genetic diversity can slow or prevent adaptation to environmental change[47] and populations that have lost diversity can become inbred, reducing fitness[48–50]. Preserving diversity before it is lost requires some assessment of the amount and partitioning of genetic variation[51].

**Table 2 | Four candidate genes in three regions of high divergence between two pairs of dark and light seed coat colour populations of Kersting's groundnut**

| Region (chr: position in MB) | Candidate gene | Enzyme/Protein | Potential link to seed colour |
|---|---|---|---|
| 2: 12.70–12.90 | *Mgeo_005191* | Cyanidin 3-O-galactoside 2″-O-xylosyltransferase | Anthocyanin production, e.g. red colouration in some kiwi[94] |
| 7: 5.10–5.25 | *Mgeo_018029* | Chalcone--flavonone isomerase-like | Flavonoid (anthocyanin) production, e.g. in mulberry[95] |
| 10: 13.20–13.30 | *Mgeo_026764* | caffeic acid 3-O-methyltransferase | Lignin production, e.g. mutations give a brown leaf midrib in maize[96] |
| 10: 13.20–13.30 | *Mgeo_026766* | Transparent testa glabra 1 | Anthocyanin production, e.g. pigments in older *Arabidopsis* leaves and stems[97] |

In this work, we chose to investigate the genome-wide diversity of Kersting's Groundnut (*M. geocarpum*). This underutilised legume has a high-quality nutritional profile[19], and is more valuable than other legumes where it is farmed, partly because it is adapted to areas of low rainfall[18]. In some locations, Kersting's Groundnut is prized relative to other, easier-to-grow/purchase beans, contributing to its value and uniqueness as a crop. Low genetic variation in Kersting's Groundnut has been reported[24,33] and therefore it is critical to understand the partitioning of variation so that variation can be conserved.

To this end, we generated a reference genome for Kersting's Groundnut and carried out a population genomic assessment of genetic variation. The genome size (365.5 MB) is smaller than *Vigna* species (for example cowpea, Adzuki bean, mung bean) and the underutilised legumes *Lablab* and *Canavalia* spp.[26], but marginally larger than the estimated genome size of horsegram (*M. uniflorum* [Lam.] Verdc.; 343.6 MB)[52]. Half of the genome was made up of repetitive sequences (50.70%), similar to cowpea (49.5%)[42] and more than lablab (43.4%)[41]. A greater percentage of the genome was made up of gypsy elements (6.18%) than copia elements (4.48%), which was found for cowpea, but the reverse was true for lablab. The Kersting's Groundnut assembly contained 33,193 genes, about 10% more than cowpea (29,773) and lablab (30,922).

The full dataset identified about 5 M SNPs when the outgroup was included, but only 0.41 M within Kersting's Groundnut, and 0.16 M after removing those in linkage disequilibrium. Compared to similar datasets analysed in the same manner, this degree of variation was ~1/18 that of lablab and 1/8 of cowpea. Kersting's Groundnut accessions were highly inbred, as reported previously[23,24]. In this study, we were not able to include any material from Togo, so the genome-wide variation could be greater than we detect here. Togo samples are genetically intermediate to the ones we sampled[24], hence we would not expect a substantial increase in genome-wide diversity if we included these. We also note that accessions chosen for sequencing span the morphological variation in the species (see 'Methods'). We assume, therefore, that Kersting's groundnut is either genetically depauperate because of a strong genetic bottleneck, a loss of varieties recently, or, perhaps more likely, Kersting's groundnut is derived from a wild taxon which is, itself, low in genetic variation. This is indeed the case, wild Kersting's Groundnut is not a common species, and has low genetic variation[33].

The population genomic analysis demonstrated accessions grouped by country, with groups of accessions sharing broadly the same seed colour, as reported based on ~900 SNP markers[24]. This grouping allowed us to compare the genomes for groups that shared similar seed colours, and identified 23 regions of high divergence, in which four potential candidate genes were located with functions in anthocyanin or lignin production. The common bean *P* locus[37] was not in a divergent region, and therefore, the genetic basis of seed colour in Kersting's Groundnut appears to be different from common bean.

Clearly, genomic diversity is very low in Kersting's Groundnut, and a conservation strategy should be employed to preserve existing germplasm and to collect as widely as possible. Given the geographic structuring of Kersting's Groundnut, we recommend a wide geographic sampling to ensure sampling as much genetic variation as possible. Strong geographic structuring likely means that varieties are locally adapted and/or preferred, which would be important to note in future breeding attempts. Strong local adaptation could mean that in situ conservation approaches are most appropriate. This may require changes to agricultural policies in Western Africa because the current policies may be hindering the conservation of underutilised crops[53]. For example, subsidies are provided for staple and or/imported crops, and other policies encourage intensification at the expense of minor crops. This, and other threats to underutilised crops such as changing cultures and removal of native habitats[54], could be putting Kersting's Groundnut under increasing pressure. Recent work to identify superior varieties has begun, and two accessions we included have been highlighted as morphologically superior (BUR8 and BUR14)[55], hence our data could be used to expedite future breeding programmes, for example if marker-assisted selection were employed[33].

The number of accessions listed in Genesys, an online global portal listing plant genetic resources for food and agriculture in seed banks, is 151, and it is not uncommon for underutilised crops to be poorly represented in seed banks[56,57]. All but two of the 151 were collected from just Nigeria and Benin; the collections therefore only represent the Eastern part of the range. Eastern accessions were also the focus of a recent GWAS, where nearly 300 accessions were studied[23]. Another investigation evaluated over 200 genotypes[24] encompassing a much broader geography, and therefore, it is imperative that these accessions are stored for future work. Further, collections of the wild taxon should also take place. Finally, the indigenous knowledge and local preferences associated with minor crops can be lost through on-farm intensification and urbanisation[58], and it is imperative to document and maintain this knowledge to ensure biodiversity, and adaptive potential are not lost[6,59].

## Methods
### Sample preparation and sequencing
TKg-36 (an accession from Nigeria), obtained from the International Institute of Tropical Agriculture (IITA), was selected for the construction of a genome assembly. Seeds were germinated in vermiculite and grown in a 1:1 mix of regular soil and vermiculite at the Chinese University of Hong Kong. Young leaves were continually collected from a single individual, immediately frozen in liquid nitrogen and kept at −80 °C until use. High-molecular-weight (HMW) DNA was extracted using the Nanobind plant nuclei kit (Pacific Biosciences, Cat No. 102-302-000) according to the manufacturer's protocol. Briefly, nuclei were isolated from 2 g of frozen leaf tissue through liquid nitrogen disruption, followed by the extraction of HMW DNA from the isolated plant nuclei using a magnetic Nanobind disk. The quality and quantity of the DNA samples were assessed with a NanoDrop spectrophotometer (Thermo Fisher Scientific) and Qubit Fluorometer

(Thermo Fisher Scientific), respectively. Library preparation and PacBio HiFi sequencing on the Sequel II platform were carried out for the DNA sample at Novogene Co., Ltd. (Beijing, China).

An Omni-C library was constructed using the Dovetail® Omni-C® Kit (Cantata Bio, Cat No. 21005) following the manufacturer's protocol. In brief, 300 mg of the frozen leaf tissue was cross-linked with 37% formaldehyde in 4 mL of 1X PBS, followed by nuclease digestion using a sequence-independent endonuclease, DNase I. Quality and quantity of lysate were assessed with a TapeStation D5000 HS ScreenTape and Qubit Fluorometer (Thermo Fisher Scientific), respectively. The qualified lysate was used for the preparation of the Omni-C library. The library was sequenced on the Illumina NovaSeq platform to generate paired-end reads of 150 bp at Novogene Co., Ltd. (Beijing, China).

Total RNA was isolated from fresh young leaf tissue using RNAiso Plus (Takara, Cat No. 9108). Briefly, 100 mg of leaf tissue was lysed with 1 mL RNAiso Plus, and chloroform was added for phase separation. The top aqueous phase containing RNA was cleaned up using the RNeasy Mini kit (Qiagen, Cat No. 74104) according to the manufacturer's protocol. The extracted RNA sample was treated with DNase I (New England Biolabs, Cat No. M0303L), followed by purification of RNA using the Monarch RNA cleanup kit (New England Biolabs, Cat No. T2040L). The quality and quantity of the RNA samples were assessed by a NanoDrop spectrophotometer (Thermo Fisher Scientific) and a Qubit Fluorometer (Thermo Fisher Scientific), respectively. A strand-specific rRNA-depleted library was constructed from the purified RNA and sequenced on the Illumina NovaSeq platform to generate paired-end reads of 150 bp at Novogene Co., Ltd. (Beijing, China). To generate long-read transcriptome data, Nanopore direct RNA sequencing was performed. Briefly, 50 µg of total RNA was enriched for poly(A) mRNA using NEBNext® poly(A) mRNA magnetic isolation module (New England Biolabs, Cat No. T2040L), followed by library preparation using Nanopore direct RNA sequencing kit (Oxford Nanopore Technologies, Cat No. SQK- RNA002) according to the manufacturer's protocol. The library was sequenced on R9.4.1 flow cell (Oxford Nanopore Technologies, Cat No. FLO-MIN106D) on the MinION system for 72 h.

## De novo genome assembly and scaffolding

De novo genome assembly of the PacBio HiFi reads was performed using Hifiasm (v 0.19.9-r616)[60] with default parameters. To further construct pseudomolecules from the PacBio assembly, Omni-C reads were first trimmed, removing adaptor sequences and bases with a quality score <20 using Trim Galore (v 0.6.7)[61]. The cleaned Omni-C reads were pre-processed with the Juicer pipeline (v 1.5.7)[62], followed by scaffolding through the 3D-DNA pipeline (v180922)[63] to correct and orient the contigs into pseudomolecules. The scaffolded assembly was visualised by JuiceBox (v 1.11.08)[64]. The completeness of the genome assembly was evaluated using BUSCO (v5.7.1)[65] using the embryophta_odb10 lineage. The LAI[35] was assessed by LTR_retriever (v3.0.1)[66]. Microsatellites were identified in the genome sequence (from just the ten chromosomes) using 'misa' (https://github.com/cfljam/SSR_marker_design/blob/master/misa.pl), identifying mono-, di-, tri-, tetra-, penta- and hexanucleotide repeats of at least 10, 8, 6, 4, 4 and 4 repeat units in length (Supplementary Data 1).

## Genome annotation

Repetitive elements, including transposable elements, simple repeats and small RNA in the scaffolded-genome assembly were identified using RepeatModeler (v 2.0)[67]. LTR retrotransposons were identified using LTR_FINDER (v 1.2)[68] and LTR_retriever (v3.0.1)[66], generating a LTR library. The de novo repeat library constructed by RepeatModeler was combined with the LTR library and RepBase (RepeatMasker-Edition version 20181026)[69] as the customised query library. The genome assembly was further soft-masked by RepeatMasker (v 4.1.0)[70] using the query library with Dfam 3.1 as the database, in which the bases within the detected repeats were converted to lowercase letters.

With the soft-masked genome, ab initio gene prediction was performed using short- and long-read RNA sequencing data. The raw RNA short reads were trimmed to remove adaptor sequences and bases with a quality score <30 using Trim Galore (v 0.6.7)[61], followed by alignment to the genome through HISAT2 (v 2.2.1)[71]. The raw Nanopore sequencing signals were pre-processed with basecalling and adaptor trimming using Guppy (v6.4.6)[72], followed by sequence alignment to the genome through Minimap2 (v 2.24-r1122)[73]. The two alignment files were merged by SAMtools (v 1.17)[74], subjected to de novo transcriptome assembly using Trinity (v 2.1.1)[75] and subsequently passed to the PASA pipeline (v 2.4.1)[76] to generate a preliminary gene annotation file. By providing Trinity-assembled transcriptome and PASA-annotated gene models as evidence, Funannotate (v 1.8.9)[77] was then used to perform ab initio gene prediction, in which it ran several gene predictors, namely Augustus, GeneMark-ES, glimmerHMM and snap, and passed all gene prediction results to Evidence Modeler to generate consensus gene models. Lastly, the untranslated regions were updated to the prediction, and gene models were fixed with RNA-seq data and Trinity-assembled transcriptome using Funannotate. Furthermore, transfer RNAs (tRNAs) were identified using tRNAscan-SE (v.2.0.11)[78] with default parameters. Long non-coding RNAs (lncRNAs) were identified by predicting the coding potential of transcripts using CPC2 (v 1.0.1)[79] and CNCI (v 2)[80] with default parameters. Transcripts longer than 200 nucleotides and labelled as 'non-coding' by either tool were retained as lncRNA. The completeness of the predicted gene set was evaluated by BUSCO (v5.7.1)[65] using the embryophta_odb10 lineage.

Functional annotation was performed using InterProScan (v5.63-95.0)[81] to assign potential functions to the protein sequences of the predicted genes by searching all available databases, including RefSeq, Pfam and PANTHER.

## Comparative genomic analysis

The 10 pseudochromosomes from Kersting's groundnut were compared with the genome of *Sphenostylis stenocarpa*[82] and *Vigna angularis* (Vigan1.1) using MCScan (v 0.9.12)[83]. Putative homologous chromosomal regions and pairwise syntenic gene blocks among the three genomes were identified, and a macrosynteny plot was constructed.

To identify gene families in Kersting's groundnut, protein sequences from the longest transcript of each gene were compared with four legumes, namely *Medicago truncatula* (MtrunA17r5.0), *Phaseolus vulgaris* (PhaVulg1_0), *Pisum sativum* (Pisum_sativum_v1a) and *Vigna angularis* (Vigan1.1) using Orthofinder (v 2.3.3)[84], using *Arabidopsis thaliana* (Araport 11) as an outgroup. With the result of gene family and a rooted, ultrametric species tree generated from Orthofinder, CAFE5 (v5.1.0)[85] was used to study expansion and contraction of the gene families in Kersting's groundnut. GO enrichment was performed with the genes in the significantly expanded and contracted gene families ($p < 0.05$) using GSEAPy (v 1.1.0)[86].

## Resequencing, phylogeny and genetic diversity

Seeds were obtained from the IITA for 10 additional accessions of Kersting's groundnut (nine from Nigeria and one from Ghana; Supplementary Data 2). Seeds were grown at the University of Southampton in a 1:1 mix of compost and vermiculite. Leaves were sampled and dried from a further 15 accessions (two from Ghana, five from Burkina Faso, eight from Benin; Supplementary Data 2). Accessions from Ghana were obtained from the University of Development Studies, and accessions from Burkina Faso were received from the Institut de l'Environnement et de Recherches Agricoles. Accessions from Benin were collected from farmers and kept at the Laboratory of Applied Ecology, University of Abomey-Calavi. Seeds were grown in open field at the Benin IITA research station. The accessions selected for sequencing were morphologically variable, for example, containing accessions with extreme (top and bottom 10%) values for the

agronomically important traits grain filling time, days to maturity, grain yield per plant and number of branches[55].

DNA was extracted from fresh or dried leaves using a modified CTAB-based protocol, treated with RNase, examined for quality, and quantified using Nanodrop and gel electrophoresis. A single accession of *M. stenophyllum* (Harms) Verdc. (NI1251 from Cameroon, obtained from Meise Botanic Garden) was grown and DNA extracted for use as an outgroup, based on its relatively close genetic relationship[87]. DNA samples were size-selected, and libraries were generated using the NEBNext® Ultra™ II FS DNA Library Prep Kit (Illumina) and NEBNext® Multiplex Oligos. Libraries were quantified, pooled and sequenced on an Illumina NovaSeqX plus by Novogene (Cambridge, UK).

Reads were trimmed using Trimmomatic (v 0.36)[88] (settings 2:30:10 LEADING:5 TRAILING:5 SLIDINGWINDOW:4:15 MINLEN:90) and only reads that remained as pairs were retained. Reads were mapped to Kersting's groundnut reference genome above (only putative chromosomes) using bowtie2 (v 2.3.1)[89] and the settings --very-sensitive-local. The resulting sam file was converted to a BAM file using samtools (v 1.20)[74] and Picard (http://broadinstitute.github.io/picard) was used to first sort the BAM file and then to identify and remove duplicate reads. The resultant files were combined into a VCF file using bcftools 'mpileup' and the bcftools commands 'call' and 'filter' used to call SNPs and to filter based on two quality ($Q > 13$ and $> 20$) and depth settings ($DP > 6$ and $> 10$). The number of SNPs retained was only slightly fewer for the strictest settings ($Q > 20$ and $DP > 10$; see results) therefore, this was used in subsequent steps.

To generate a phylogeny, initially indels, SNPs with missing data from >3 samples or with MAF < 0.05 were removed from the 27-sample VCF file using vcftools (v 0.1.16)[90]. Then PLINK (v 1.9)[91] was used to identify SNPs in LD using settings 50 5 0.5 and these were removed using vcftools. 1000 distance matrices were constructed with VCF2Dis (https://github.com/BGI-shenzhen/VCF2Dis; v 1.50) and analysed in FastME[92] to generate a NJ tree with bootstrap values. The tree was visualised in iTOL (https://itol.embl.de/) and rooted with the *M. stenophyllum* sample.

Prior to population genomic analysis, the outgroup was removed from the VCF file, and only polymorphic SNPs were retained. The same settings as above were used to remove SNPs with missing data, low MAF and those in LD. STRUCTURE ver. 2.3.4[93] was used to determine the most likely number of population clusters with five runs per K (number of clusters) comprising 20,000 iterations after a burn-in of 10,000.

Genetic diversity ($\pi$) and population divergence ($F_{ST}$) were estimated using vcftools as the average per 100 kb window (in steps of 50 kb) for the clusters of accessions identified by STRUCTURE. Genetic diversity in Kersting's groundnut was compared to that in two other African legumes, lablab and cowpea. For an accurate comparison, the same settings as above were used, on a similar number of accessions. Analysis of 26 domesticated lablab samples used eight samples from Njaci et al.[41] and 18 newly sequenced samples (NCBI BioProject PRJNA1267693). Analysis of 18 cowpea samples utilised 17 from divergent geographic locations from an NCBI BioProject (PRJNA326685 [https://www.ncbi.nlm.nih.gov/bioproject/PRJNA326685]) and reads from the sample previously assembled into the cowpea reference genome (IT97K-499-35; https://www.ncbi.nlm.nih.gov/sra/SRR19215719). Sample details are given in Supplementary Data 3.

### Candidate genes for seed colour
Kersting's groundnut accessions are primarily identified by seed colour, and the clusters of samples we recovered from the population genomic analysis corresponded to seed colour clusters (see results). We therefore reasoned that highly divergent regions between sister clusters that differed in seed colour could contain loci involved in this important morphological difference. Several other traits are likely to differ between sister clusters (e.g. those underlying local adaptation) therefore, we carried out two comparisons between separate light and dark-seeded clusters and cross-checked the results, focussing on highly divergent regions resolved in both comparisons. We used $F_{ST}$ as a metric for divergence between clusters and identified windows of SNPs containing SNPs in the top 5% and 1% for $F_{ST}$ (calculated above). Windows in the top 5% in both comparisons were identified, and the surrounding region was extracted and searched for genes putatively involved in pigments or anthocyanins. We also identified a putative orthologue of the known seed colour gene *P* from *Phaseolus vulgaris*[37] in the Kersting's groundnut genome.

### Reporting summary
Further information on research design is available in the Nature Portfolio Reporting Summary linked to this article.

## Data availability
PacBio, Omni-C, Illumina RNA sequencing and Nanopore direct RNA sequencing data have been deposited in the NCBI Sequence Read Archive under BioProject PRJNA1173951. These data and the genome assembly have been deposited in the Genome Warehouse in the National Genomics Data Center, Beijing Institute of Genomics, Chinese Academy of Sciences/China National Center for Bioinformation under BioProject PRJCA038461. The genome assembly and gene annotation files of Kersting's groundnut have been additionally deposited in Figshare [https://doi.org/10.6084/m9.figshare.28783517.v1] and the CUHK data repository [https://doi.org/10.48668/57WB4O]. Kersting's Groundnut and lablab resequencing data in fastq format are available at the NCBI Sequence Read Archive under accession PRJNA1185675 and PRJNA1267693. The VCF containing the reads aligned to the genome is available at Figshare [https://doi.org/10.6084/m9.figshare.29100578.v1]. Source data are provided with this paper.

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

## Acknowledgements
T.-F.C. and M.A.C. were funded in part through an institutional award to the University of Southampton from the BBSRC (Grant Ref: BB/X512035/1). M.A.C. was funded to visit the lab of T.-F.C. through the Kan Tong Po Visiting Fellowships programme of the Royal Society (Grant Refs: KTP\R1\211008 and KTP\R1\231030). M.A.C. acknowledges the use of the IRIDIS High Performance Computing Facility and associated support services at the University of Southampton. We thank IITA (International Institute of Tropical Agriculture), especially Emily Iwu, for the seed of Kersting's Groundnut and Meise Botanic Garden and other seed banks named in the 'Methods' section for additional seed. T.-Y.C and T.-F.C. were in part supported by the National Natural Science Foundation of China (NSFC)/Research Grants Council (RGC) of Hong Kong Joint Research Scheme N_CUHK488/22, the Hong Kong Research Grants Council Area of Excellence Scheme (AoE/M-403/16), the General Research Fund (14100422), a donation from Mr and Mrs Sunny Yang, and the Innovation and Technology Commission, Hong Kong Special Administrative Region Government to the State Key Laboratory of Agrobiotechnology (The Chinese University of Hong Kong). The opinions, findings, conclusions or recommendations expressed in this publication do not reflect the views of the Government of the Hong Kong Special Administrative Region or the Innovation and Technology Commission. The funders had no role in the study design, data collection and interpretation, or the decision to submit the work for publication.

## Author contributions
M.A.C. and T.-F.C. conceived and planned the experiments. T.-Y.C. and T.-F.C. performed DNA extraction and PacBio Sequencing for the reference genome, and T.-Y.C. performed the genome assembly and annotation. M.A.C., K.M.K. and E.E.A. carried out plant growth and DNA extraction for the resequencing samples using phenotype data from K.M.K. and E.E.A. M.A.C. analysed the resequencing data, performed the diversity analyses and compared the groups that differed in seed colour. M.A.C. and T.-Y.C. wrote the original draft of the manuscript. All authors reviewed and approved the final version of the manuscript.

## Competing interests
The authors declare no competing interests.

## Ethics approval
Local researchers have been included in the research process where possible, and all researchers who provided personally collected material are included as authors. Other samples come from recognised seed banks. All data are freely available to any researcher, including those from where the samples were collected. Biological materials (leaves and seeds) have been sent from their origin to the UK for DNA sequencing, but are available from K.M.K. and E.A.A. at the University of Abomey-Calavi, Benin (leaves) or the seed banks (seeds).
