## [Peer Review file · Nature Communications]

Exceptionally low genomic diversity in the underutilised legume Kersting's groundnut (*Macrotyloma geocarpum* (Harms) Maréchal & Baudet)

Corresponding Author: Professor Mark Chapman

Version 0:

Reviewer comments:

Reviewer #1

(Remarks to the Author)

This is an interesting work. The authors have done an excellent job in terms of sequencing the whole genome of Kersting's 1 groundnut (*Macrotyloma geocarpum*) with long read sequencing using Pac Bio platform. The statistics of the sequencing is very convincing. Based on the comparative genomics they achieved by comparing the genome sequences of other related leguminous species like *Phaseolus* and *Vigna*. They were able to map around 83% of the scaffolds on to 10 pseudochromosomes. The major inference from their sequencing of the Kersting's groundnut was to show a very less molecular diversity. Also, the resequencing of a panel of accessions of Kersting's groundnuts taken from various geographical locations also showed less variability.

While this work is fine, the major premise of their investigation was to suggest the conservation of this important crop species, which is largely classified as an "under-utilized" plant. Though the authors indicate the importance of this crop, there has been little effort in terms of collating the germplasm and characterizing the available lines for Agronomic or Physiological traits, except some traits like seed coat colour.

I agree that these under-utilized crop plants hold a tremendous potential in our future efforts towards sustainability of food and nutrient security. Unfortunately, the genome sequencing does not indicate any strategies of how we can preserve and utilize the values of these crops. Moreover, the existence of low diversity in phenotypic characters as well as the genome gives us very less options to utilise this crop as future foods.

It would be beneficial had some systematic work is initiated to screen all available germplasm for diversity in many agronomic traits that may hold the key for the success of this crop as future food or to serve as a repository of useful traits that can be used for improving other related crop species.

At this stage is a mere documentation of whole genome sequencing which has narrow applicability at this juncture. I am not enthusiastic to accept this manuscript.

Reviewer #2

(Remarks to the Author)

Reviewer #3

(Remarks to the Author)

Summary of the key results

The manuscript titled "Exceptionally low genomic diversity in the underutilised legume, Kersting's groundnut (*Macrotyloma*

geocarpum (Harms) Maréchal & Baudet” by Cheung and colleagues presents the development of the first reference genome for Kersting’s Groundnut, offering valuable insights into its genetic diversity, population structure, and conservation priorities. Through the resequencing of a diverse panel of accessions, the study revealed that accessions are grouped based on geographic origin and seed coat colour. The findings highlight exceptionally low genetic diversity and high levels of inbreeding in the crop, with genetic diversity estimated to be 85–95% lower compared to lablab and cowpea. Additionally, the study identified four candidate genes associated with seed coat colour, linked to pathways involved in pigment and flavonoid production.

Originality and significance:

The study is novel in presenting the first reference genome and population genomic analysis for Kersting’s groundnut. While previous works have uncovered genetic diversity using marker-based approaches, this study adds a genomic dimension that is critical for conservation and breeding strategies.

Data & methodology:

Line 118: The scaffold number increased from 1044 contigs to 1374 scaffolds after incorporation of Omni-C data, which is counterintuitive since scaffolding usually reduces the number of scaffolds. Please clarify why the scaffold count increased post Omni-C data and provide more information regarding the polishing process to address this irregularity.

Line 123: A large proportion (22.74%) of repetitive sequences were unclassified. Please provide more information on efforts to classify these repetitive sequences, such as using alternative tools or databases.

Line 144: Only two legume genomes (common bean and adzuki bean) were used for syntenic comparison in this study. The authors can expand comparisons by including more closely related legumes with similar genome architecture.

Lines 146-148: Although the synteny is conserved, no further details are provided on the nature or potential causes of inter-chromosomal rearrangements. Please include a brief description of evolutionary or biological consequences of these inter-chromosomal rearrangements.

Lines 164-166: Expanded gene families were mainly enriched for terms related to photosynthesis and energy production, but their connection to Kersting’s groundnut biology is not explored. Please discuss why photosynthesis and energy production-related terms are important and their potential roles in adaptation or growth.

Appropriate use of statistics and treatment of uncertainties

Lines 140-142: The high BUSCO scores are promising but should be complemented with additional metrics, such as LTR assembly index (LAI) or k-mer based completeness.

Line 258: The manuscript mentions the absence of Togo samples but does not adequately quantify or discuss the potential bias introduced.

Line 267: The interpretation of low diversity should include broader discussion of possible sampling effects or historical bottlenecks.

Conclusions:

Lines 267-268: While the need for conservation is pointed out, specific actionable strategies are lacking in the manuscript. For instance, how can genetic diversity of Kersting’s groundnut be preserved given the low diversity and geographic structuring? The authors can comment on more detailed conservation measures, such as cryopreservation, or in situ conservation, among others.

Line 229-281: The discussion highlights similarities and differences with other underutilized legume crops but misses the opportunity to draw broader conclusions about the uniqueness of Kersting’s groundnut.

Suggested improvements:

Lines 250-252: The higher percentage of Gypsy elements in Kersting’s groundnut compared to lablab but the reverse trend observed for Copia elements is mentioned but not interpreted biologically or evolutionarily. Please discuss in detail how these findings may relate to the evolution or adaptation of Kersting’s groundnut compared to related legumes.

Line 226: Potential pigmentation-related genes associated with seed coat colour were identified, but their roles remain speculative. Furthermore, there are no experimental validations (e.g., gene knockouts, overexpression studies, phenotypic assays) to confirm the functional roles of these genes in controlling seed coat colour in Kersting’s groundnut. Without such validation, the proposed functions and evolutionary significance of these genes remain speculative.

Lines 255-257: The ‘Discussion’ section briefly highlights the low genetic variation (~1/18th of lablab and 1/8th of cowpea) in Kersting’s groundnut but does not examine into the potential causes, such as domestication bottlenecks, restricted geographical range, or limited gene flow. Please include an analysis to confirm whether these factors have contributed to the observed diversity patterns.

References:

The manuscript gives appropriate credit to previous work, but the integration of references in key sections could be improved. For example:

Lines 245-253: Broader context on genome evolution in legumes should be incorporated when discussing repetitive elements and genome size.

Lines 257-260: The authors mention the absence of samples from Togo and suggest that including them might not significantly change genome-wide diversity estimates. However, this point is not substantiated or elaborated. Please clarify this assumption with references or additional data and discuss the limitations of this omission.

Clarity and context:

The abstract efficiently summarizes the main findings but could highlight the importance of low genetic diversity and conservation strategies more prominently.

Lines 51-58: The advantages of underutilized crops are discussed widely but can include more precise examples or comparative insights. For example, highlighting key underutilized crops that have successfully transitioned to mainstream agriculture with the help of genomic interventions would offer deeper justification.

Lines 87-92: Instead of simply listing challenges such as disease susceptibility or longer cooking times, please outline how genomic studies could help overcome these problems, for instance, by identifying resistance loci or improving processing traits.

Lines 252-253: The 10% higher gene count in Kersting's groundnut compared to lablab and cowpea is documented but not discussed. The authors can explore potential reasons for this difference, such as gene family expansion, and link it to key agronomic traits or adaptation.

Line 261: Although the clustering of diverse accessions by country and seed colour is mentioned, there is no investigation of its agricultural, cultural, or evolutionary implications. The authors can discuss how geographic structuring influences Kersting's groundnut breeding or conservation strategies.

Version 1:

Reviewer comments:

Reviewer #1

(Remarks to the Author)

After carefully going through the revision of the manuscript and the answers to queries from reviewers, the Manuscript looks better. The authors have included more accessions for phenotyping and the diversity is better explained now. Their explanation of the traits that would be useful in further breeding to improve the species is not very convincing. Days to maturity, color of seeds and such traits have little influence on productivity. If an underutilized species should develop into sustaining food and nutrient security, it is extremely important that "traits" that determine yield under field conditions should be determined by phenotyping.

Generating genomic information and the subsequent analysis of genomics data is well done. It would be better if they can provide information and create a database for the SNP and SSR markers from the sequencing analysis, which would strongly enhance the translatability of the work towards crop improvement.

Reviewer #2

(Remarks to the Author)

Reviewer #3

(Remarks to the Author)

I have reviewed the revised manuscript and find that the authors have satisfactorily addressed the comments and concerns raised during the initial review. They have provided well-reasoned responses and incorporated necessary revisions, strengthening the manuscript's clarity, methodology, and overall impact. Based on these improvements, I recommend the manuscript for publication.

We thank the reviewers for their constructive comments on our submission which we have taken on board for our revised version. Note that in the reanalysis some numbers have changed in the manuscript, although all are minor (but note the improvement in the TE annotation which was requested by reviewer 3). Line numbers referred to below are in the track changes version.

Reviewer 1.

1.1. It would be beneficial had some systematic work is initiated to screen all available germplasm for diversity in many agronomic traits that may hold the key for the success of this crop as future food or to serve as a repository of useful traits that can be used for improving other related crop species. At this stage is a mere documentation of whole genome sequencing which has narrow applicability at this juncture. I am not enthusiastic to accept this manuscript.

Previous analyses of the species have examined morphological variation in broad sets of germplasm (<https://doi.org/10.1371/journal.pone.0234769> and <https://assets-eu.researchsquare.com/files/rs-4831288/v1/06e44414-3649-4561-9e40-3bb8cb94d485.pdf?c=1725906066>), therefore this has already been carried out.

Notably, 13 of our accessions are among the 80 considered in the latter study and we selected these 13 to encompass a range of morphologies. For example, our sequenced accessions from Benin and Burkina Faso included accessions from both the top and bottom 10% of the agronomically important traits 'grain filling duration', 'days to 50% maturity', 'grain yield per plant', 'number of branches', and 'seed thickness'. We also include two accessions (BUR8 and BUR14) which were noted in that study as having superior morphologies therefore may be used in breeding programmes going forward. This highlights the use of our data beyond the existing study. We acknowledge this justification was missing from the first submission and it should have been included.

We now highlight the above points in the manuscript as follows:

Line 100: We now point out that large collections have been phenotyped.

Lines 204-5 and 274-5: we emphasise this was a panel of geographically and morphologically diverse accessions.

Lines 411-3: we give more information on the lines we selected being morphologically variable.

Reviewer 2.

Co-reviewed with 1 or 3

Reviewer 3 (please note, we have reordered the comments to represent the order in which they appear in the manuscript).

3.1. Lines 51-58: The advantages of underutilized crops are discussed widely but can include more precise examples or comparative insights. For example, highlighting key underutilized crops that have successfully transitioned to mainstream agriculture with the help of genomic interventions would offer deeper justification.

We do not have evidence that genomic interventions have elevated specific crops, except for coincident observations; however, we agree that some previously minor crops are now more mainstream, which we have added (lines 66-67).

3.2. Lines 87-92: Instead of simply listing challenges such as disease susceptibility or longer cooking times, please outline how genomic studies could help overcome these problems, for instance, by identifying resistance loci or improving processing traits.

This is a good point; we have added a sentence to improve this (lines 93-95)

3.3. Line 118: The scaffold number increased from 1044 contigs to 1374 scaffolds after incorporation of Omni-C data, which is counterintuitive since scaffolding usually reduces the number of scaffolds. Please clarify why the scaffold count increased post Omni-C data and provide more information regarding the polishing process to address this irregularity.

Omni-C scaffolding provides spatial information on chromatin interactions, which helps correct misassemblies and rearrange contigs accordingly. However, this process can fragment larger contigs into smaller ones, should there be an initial misassemble. Hence, this is what was observed here.

After assembling the PacBio HiFi reads, we observed 10 contigs exceeding 20 Mb in length, despite a total of 1,044 contigs. Given the high contiguity and N50 of our contig-level assembly, Omni-C scaffolding resulted in only a marginal improvement to those 10 large contigs while increasing the scaffold count. Although incorporating Omni-C data improved the accuracy of the genome assembly in terms of structural and spatial organisation by splitting and re-orientating contigs, it also led to a higher scaffold count. This adjustment enhances the reliability of our assembly. We have clarified this point in the results section (Lines 122-5).

3.4. Line 123: A large proportion (22.74%) of repetitive sequences were unclassified. Please provide more information on efforts to classify these repetitive sequences, such as using alternative tools or databases.

To further classify the unknown repetitive sequences, we expanded our analysis by utilizing additional tools and databases. We attempted three approaches, one of which improved the classification.

The approach which improved the classification used RepeatModeler to generate a de novo repeat library and combined it with an LTR retrotransposon library created using LTR_FINDER and LTR_retriever. These two libraries were then merged with the RepBase library to create a customised query library for RepeatMasker. RepeatMasker, using this customized library along with Dfam 3.1 database, identified and classified repetitive elements in the genome. With this workflow, the unclassified repeats reduced from 83,077,495 bp (22.74%) to 72,150,883 bp (19.74%). Incorporating the LTR library substantially increased the classified LTR elements from 43.7 Mbp (11.98%) to 65.0 Mbp (17.81%), while other repetitive elements showed only marginal changes. Details of the tools and database used have been added to the methodology (Lines 364-8), and the results (Lines 128-138) have been revised accordingly. Figure 1c and Table 1 were also updated to reflect these findings.

Two other pipelines were tested, Earl Grey (Transposable Element Annotation and Analysis Pipeline)¹ and RED (REpeat Detector)². Both left a substantial portion of repetitive sequences unclassified. Earl Grey identified 106.9 Mbp (29.24%) as unclassified, while RED detected only 125.0 Mbp (34.20%) of repetitive sequences in total, indicating that RED is even less effective than the other methods.

Overall, while 19.74% of repeats remained unclassified using our RepeatModeler workflow, this represents the best result we were able to achieve.

¹ Baril, T., Galbraith, J., & Hayward, A. (2024). Earl Grey: A Fully Automated User-Friendly Transposable Element Annotation and Analysis Pipeline. *Molecular biology and evolution*, 41(4), msae068. <https://doi.org/10.1093/molbev/msae068>

² Girgis H. Z. (2015). Red: an intelligent, rapid, accurate tool for detecting repeats de-novo on the genomic scale. *BMC bioinformatics*, 16, 227. <https://doi.org/10.1186/s12859-015-0654-5>

3.5. Line 144: Only two legume genomes (common bean and adzuki bean) were used for syntenic comparison in this study. The authors can expand comparisons by including more closely related legumes with similar genome architecture.

We were aware of this, but given the phylogenetic placement of Kersting's Groundnut (<https://doi.org/10.3732/ajb.1100069>), the closest relatives with genome sequences available at the time of submission were collectively in the genera Lablab, Vigna and Phaseolus, hence adding more species would only give us equally- or more distantly-related species, not closer as the reviewer asks. The only exception is horsegram, for which the required files are not publicly available.

However, since submission, the African Yam Bean genome has been published (<https://doi.org/10.1038/s41597-024-04210-2>), therefore we have added this to the analysis. A macrosynteny plot between African yam bean, Kersting's Groundnut and adzuki bean (Figure 2) has been generated accordingly. The necessary text has been edited too (lines 152-3, 391-2).

3.6. Lines 140-142: The high BUSCO scores are promising but should be complemented with additional metrics, such as LTR assembly index (LAI) or k-mer based completeness.

We have now incorporated LAI alongside BUSCO scores to provide a more comprehensive assessment of the genome assembly quality. An overall LAI value of 17.2 was reported, which indicates a reference-grade genome. The main text (Lines 148-151) and methods (line 360) have been updated to include this information, and an LAI distribution plot has been added as Figure 1e.

3.7. Lines 146-148: Although the synteny is conserved, no further details are provided on the nature or potential causes of inter-chromosomal rearrangements. Please include a brief description of evolutionary or biological consequences of these inter-chromosomal rearrangements.

Rearrangements are a process of evolutionary divergence, and the cause is simply time, we cannot speculate anything else. Trying to tie these to any consequence would not be appropriate or possible. We agree that some description is warranted and so describe the key changes related to chromosome number (lines 157-161).

3.8. Lines 164-166: Expanded gene families were mainly enriched for terms related to photosynthesis and energy production, but their connection to Kersting's groundnut biology is not explored. Please discuss why photosynthesis and energy production-related terms are important and their potential roles in adaptation or growth.

As this is not a comparison of Kersting's groundnut with a closely related species (for example the wild progenitor), we are essentially comparing two divergent lineages each with hundreds of species. Any supposition related to adaptation in Kersting's groundnut itself would be incorrect.

3.9. Line 226: Potential pigmentation-related genes associated with seed coat colour were identified, but their roles remain speculative. Furthermore, there are no experimental validations (e.g., gene knockouts, overexpression studies, phenotypic assays) to confirm the functional roles of these genes in controlling seed coat colour in Kersting's groundnut. Without such validation, the proposed functions and evolutionary significance of these genes remain speculative.

We are very open to the fact that these are candidate genes and in no way are we saying causative, for example by saying "candidate" in the subheading and "potential roles in pigmentation" at the end of the section.

3.10. Lines 245-253: Broader context on genome evolution in legumes should be incorporated when discussing repetitive elements and genome size.

We are not sure what the reviewer would like us to add here. Our paper is not about legume genome evolution, and we have compared the LTRs, genome size, gene number, and synteny to

closely related species already in this paper. To compare Kersting's groundnut to many other species seems unnecessary.

3.11. Lines 250-252: The higher percentage of Gypsy elements in Kersting's groundnut compared to lablab but the reverse trend observed for Copia elements is mentioned but not interpreted biologically or evolutionarily. Please discuss in detail how these findings may relate to the evolution or adaptation of Kersting's groundnut compared to related legumes.

As for the point above related to gene family expansions and contractions (3.8), this would be comparing the Kersting's groundnut lineage to the lablab lineage and hence we cannot tie this in any way to the biology of Kersting's groundnut, and it would be incorrect to try to.

3.12. Line 229-281: The discussion highlights similarities and differences with other underutilized legume crops but misses the opportunity to draw broader conclusions about the uniqueness of Kersting's groundnut.

This is covered in the introduction and largely reiterated at the start of the discussion. We take on board the point about comparing to other crops, hence we have added a sentence to explain how Kersting's groundnut is prized in certain communities over other local beans (lines 255-7).

3.13. Lines 252-253: The 10% higher gene count in Kersting's groundnut compared to lablab and cowpea is documented but not discussed. The authors can explore potential reasons for this difference, such as gene family expansion, and link it to key agronomic traits or adaptation.

See points 3.8 and 3.11 where we believe that tying differences in gene family size, gene count, or TE content to adaptation in Kersting's Groundnut is inappropriate given the divergent species being compared.

3.14. Lines 257-260: The authors mention the absence of samples from Togo and suggest that including them might not significantly change genome-wide diversity estimates. However, this point is not substantiated or elaborated. Please clarify this assumption with references or additional data and discuss the limitations of this omission.

And

3.15. Line 258: The manuscript mentions the absence of Togo samples but does not adequately quantify or discuss the potential bias introduced.

Previous research (<https://doi.org/10.1371/journal.pone.0234769>) demonstrates there is greater genetic differences between agro-ecological regions within Benin than between Benin and Togo. This means that the absence of Togo samples is unlikely to affect our overall findings of drastically low genetic diversity in Kersting's Groundnut. We discuss this already by stating "Togo samples are genetically intermediate to the ones we sampled²⁴, hence we would not expect a substantial increase in genomewide diversity if we included these." (lines 273-4)

3.16. Lines 255-257: The 'Discussion' section briefly highlights the low genetic variation (~1/18th of lablab and 1/8th of cowpea) in Kersting's groundnut but does not examine into the potential causes, such as domestication bottlenecks, restricted geographical range, or limited gene flow. Please include an analysis to confirm whether these factors have contributed to the observed diversity patterns.

And

3.17. Line 267: The interpretation of low diversity should include broader discussion of possible sampling effects or historical bottlenecks.

We have now two sentences hypothesising about this (lines 275-9). We are not aware of any formal analysis that could be used to tease apart these hypotheses, however.

3.18. Lines 267-268: While the need for conservation is pointed out, specific actionable strategies are lacking in the manuscript. For instance, how can genetic diversity of Kersting's groundnut be preserved given the low diversity and geographic structuring? The authors can comment on more detailed conservation measures, such as cryopreservation, or in situ conservation, among others.

We agree this section reads as incomplete. We have added this now (lines 289-92, 296-8, 305-308).

3.19. Line 261: Although the clustering of diverse accessions by country and seed colour is mentioned, there is no investigation of its agricultural, cultural, or evolutionary implications. The authors can discuss how geographic structuring influences Kersting's groundnut breeding or conservation strategies.

We have added some more text to explain how this arose and why it is significant to know this for future breeding of the crop (lines 289-90, 298 and 306).

Here we provide our response to the reviewers comments on our resubmitted manuscript. We thank the editor and reviewers for their comments.

Reviewer #1 (Remarks to the Author):

“Their explanation of the traits that would be useful in further breeding to improve the species is not very convincing. Days to maturity, color of seeds and such traits have little influence on productivity. If an underutilized species should develop into sustaining food and nutrient security, it is extremely important that "traits" that determine yield under field conditions should be determined by phenotyping.”

We agree that yield is usually the most valuable trait, and we have referenced a preprint where yield has been assayed. In addition, we used yield phenotypes (among others) to decide which accessions to include in the resequencing analysis. This is explained in the methods.

“Generating genomic information and the subsequent analysis of genomics data is well done. It would be better if they can provide information and create a database for the SNP and SSR markers from the sequencing analysis, which would strongly enhance the translatability of the work towards crop improvement.”

This is a valuable suggestion; we have submitted the SNP and indel variants to figshare as a VCF and carried out an analysis of SSRs (additional text added to the results and methods, and a new supplementary file added.

Reviewer #2 (Remarks to the Author):

“I co-reviewed this manuscript with one of the reviewers who provided the listed reports. This is part of the Nature Communications initiative to facilitate training in peer review and to provide appropriate recognition for Early Career Researchers who co-review manuscripts.”

No response required.

Reviewer #3 (Remarks to the Author):

“I have reviewed the revised manuscript and find that the authors have satisfactorily addressed the comments and concerns raised during the initial review. They have provided well-reasoned responses and incorporated necessary revisions, strengthening the manuscript’s clarity, methodology, and overall impact. Based on these improvements, I recommend the manuscript for publication.”

No response required.